# Fairness matters for change: A multilevel study on organizational change fairness, proactive motivation, and change-oriented OCB

**Bin Ling**[1], **Qu Yao**[1], **Yutong Liu**[1,2]*, **Dusheng Chen**[3]

**1** Business School, Hohai University, Nanjing, China, **2** Department of Psychology, Harbin Normal University, Harbin, China, **3** Hangzhou Hikvision Digital Technology Co., Ltd., Hangzhou, China

* liuyutong1214@163.com

**Data Availability Statement:** The data underlying the results presented in this study are publicly available in the Open Science Framework (OSF) repository. The data can be accessed at the

## Abstract

The success of organizational change often hinges on the perception of fairness within a change unit. This group-level organizational change fairness is crucial for enhancing proactive motivation states and fostering positive change-oriented organizational citizenship behavior (OCB). Rooted in the proactive motivation model, this study establishes a comprehensive multilevel framework to investigate the influence of group-level organizational change fairness on employees' change-oriented OCB. It explores the mediating role of three proactive motivational states and the moderating impact of perceived change impact. Analyzing data collected from 597 employees within 107 teams across 43 Chinese companies, our findings indicate that group-level perceived organizational change fairness significantly predicts employees' change-oriented OCB through organizational change self-efficacy, involvement, and positive emotional experiences. Furthermore, the study reveals that group-level perceived change impact moderates the relationship between group-level fairness perception and both change self-efficacy and positive emotional experiences, with stronger associations observed under conditions of low perceived change impact. These insights notably advance our understanding of the cross-level determinants influencing change-oriented OCB through perceived fairness and proactive motivation. Managers should focus on developing fairness perceptions to stimulate OCB by fostering employees' proactive motivation states, particularly during low-impact organizational changes. Our findings provide valuable implications for organizational change management practices.

## Introduction

Today, organizational change has evolved into an integral facet of modern business practices, characterized by ongoing transformations in structures, processes, and cultures [1]. The success of organizational change often hinges not only on their implementation but also on how fairly they are perceived by employees [2, 3]. Organizational change fairness holds substantial

following URL: https://osf.io/5xcun/?view_only=d29a262523c2444ab024d224f4e33579.

**Funding:** This research was supported by Soft Science Research Plan Project for Nanjing (202303001).

significance as it impacts how employees behave during the process of change within the organization. For instance, when a leader demonstrates procedural fairness during organizational change, employees are more willing to accept the change [4]. Although a few studies show that organizational change fairness predicts work outcomes [4, 5], little is known about how organizational change fairness shape employee change-oriented OCB at work, a sought-after yet demanding behavior within today's organizations.

Change-oriented OCB refers to "constructive efforts by individuals to identify and implement changes concerning work methods, policies, and procedures to improve the situation and performance" [6]. This behavior is highly desired by organizational because it is helpful to improve organizational development and the possibility of survival from organizational change [6–8]. Research has consistently shown that change-oriented OCB is a type of change-related proactive behavior driving organizational change [8] and has been associated with a range of organizational outcomes, such as turnover intention and career commitment [9], overall job performance [10], and organizational commitment [11]. Therefore, understanding how to promote the endurance of this behavior within organizational change is crucial but underexplored.

While research has extensively examined OCB and organizational change separately, there is a discernible fragmentation in understanding OCB's role during change initiatives. This knowledge gap poses a problem as the successful execution of change relies, in part, on the voluntary actions and proactive responses of employees to the change process. Furthermore, prevailing studies examining change-oriented OCB have predominantly focused on individual attributes (such as adaptability, traits, and tenure) and situational factors (such as supportive, transformational, and empowering leadership) within the confines of a stable work environment [6–8, 12]. Examining change-oriented OCB mainly in a stable work environment may limit the understanding of how these behaviors manifest or adapt during periods of organizational change, potentially leading to incomplete insights into the factors that drive such behaviors in dynamic work settings. Thus, exploring predictive mechanisms of change-oriented OCB necessitates a multi-level analysis approach that considers both organizational and individual factors, such as organizational change fairness.

In the context of organizational change, studying the relationship between fairness, motivation, and OCB is essential because organizational change fairness can be a key driver of change-oriented OCB. Understanding how organizational change fairness influences employees' motivational states allows us to better explain their behaviors during organizational change. It is necessary to use the proactive motivation model [13] as the theoretical framework to explain the relationship between organizational change fairness and change-oriented OCB because this model systematically elucidates how motivational states influence individual proactive behaviors. The proactive motivation model identifies three motivational states—"can do", "reason to", and "energized to" [13]—that drive individual behavior and underpin engagement in proactive behaviors such as personal initiative and OCB [14, 15]. These motivational states are crucial for understanding how organizational change fairness influences employees' motivation, particularly in response to change initiatives.

Specifically, perceived fairness can stimulate employees' proactive tendencies, fostering change-oriented OCB by providing insights into how fairness perceptions motivate employees to proactively support organizational change efforts. The proactive motivation model suggests that these motivational states are shaped by direct experiences of the change. Positive experiences with change provide employees with a "reason to" engage, as they see beneficial outcomes and are thus more likely to support the change. Furthermore, change involvement relates to the "can do" and "energized to" states, as active participation in the change process enhances employees' sense of control and commitment. Change self-efficacy, meanwhile,

strengthens the "can do" belief, as employees feel capable of overcoming challenges associated with the change. By incorporating these mediating factors, especially distinguishing between aspects at the team and individual levels, our study recognizes how organizational change fairness across these levels energizes employees' motivational states, impacting their change-oriented OCB. Additionally, drawing from prior research, the activation of proactive states could be influenced by specific situational factors [13, 16]. Therefore, in formulating our theoretical framework, we must account for potential boundary conditions. Our study identifies change impact as a key boundary factor, suggesting that it might either amplify or hinder employees' motivational states during the change process.

In summary, this study examines how organizational change fairness affects individuals' change-oriented OCB by influencing their motivational states, potentially impacted by change impact. Our constructed model (Fig 1) aims to elucidate the mechanisms behind change-oriented OCB within an organizational change context.

## Theoretical foundation and model development

### Group-level change fairness and change-oriented OCB

In today's rapidly changing business environment, organizational change is crucial for corporate success. The successful execution of change relies on employees, especially those demonstrating change-oriented OCB [8]. Change-oriented OCB aims to enhance organizational effectiveness and is influenced by various contextual factors, among which organizational change fairness profoundly impacts employee responses and behaviors [2]. However, research on promoting change-oriented OCB during organizational change is limited. This study explores the impact of organizational change fairness on change-oriented OCB to address this gap.

Initially, organizational change fairness was linked to broader assessments of procedural justice [17]. As research progressed, other aspects like organizational fairness were identified [18]. Notably, discussions about justice among team members often lead to increasingly aligned perceptions of organizational fairness [19], indicating organizational change fairness forms a collective perception through individual-organizational interactions. Thus, this study focuses on organizational change fairness as a group-level construct.

Several studies have demonstrated that organizational change fairness significantly impacts positive employee attitudes and behaviors during periods of change [2]. When change is perceived as being managed fairly, the implementation process tends to proceed more smoothly [20]. Assessments of organizational change fairness emerge from shared information, an enhanced understanding of change, and the cultivation of positive, equitable relationships. These factors aid in transforming employee attitudes toward change and fostering engagement [21]. Parker et al.'s proactive motivation model provides insight into the connection between organizational change fairness and change-oriented OCB [13]. According to this model, proactive individuals are relatively less influenced by environmental constraints. Instead, they take the initiative to drive environmental change by identifying opportunities, displaying initiative, acting, and persisting in their efforts, such as through job crafting [22]. Within the context of organizational change fairness, there is an increase in opportunities for employee involvement and participation, resulting in positive motivation and perceived control [23, 24]. Consequently, employees can more effectively adapt to changing demands by exercising greater control over these demands, utilizing information to guide their behaviors, and leveraging their heightened involvement.

Furthermore, organizational fairness is crucial for promoting OCB, where a climate of procedural fairness boosts positive attitudes, behaviors, and outcomes, motivating OCB [5]. Yu's

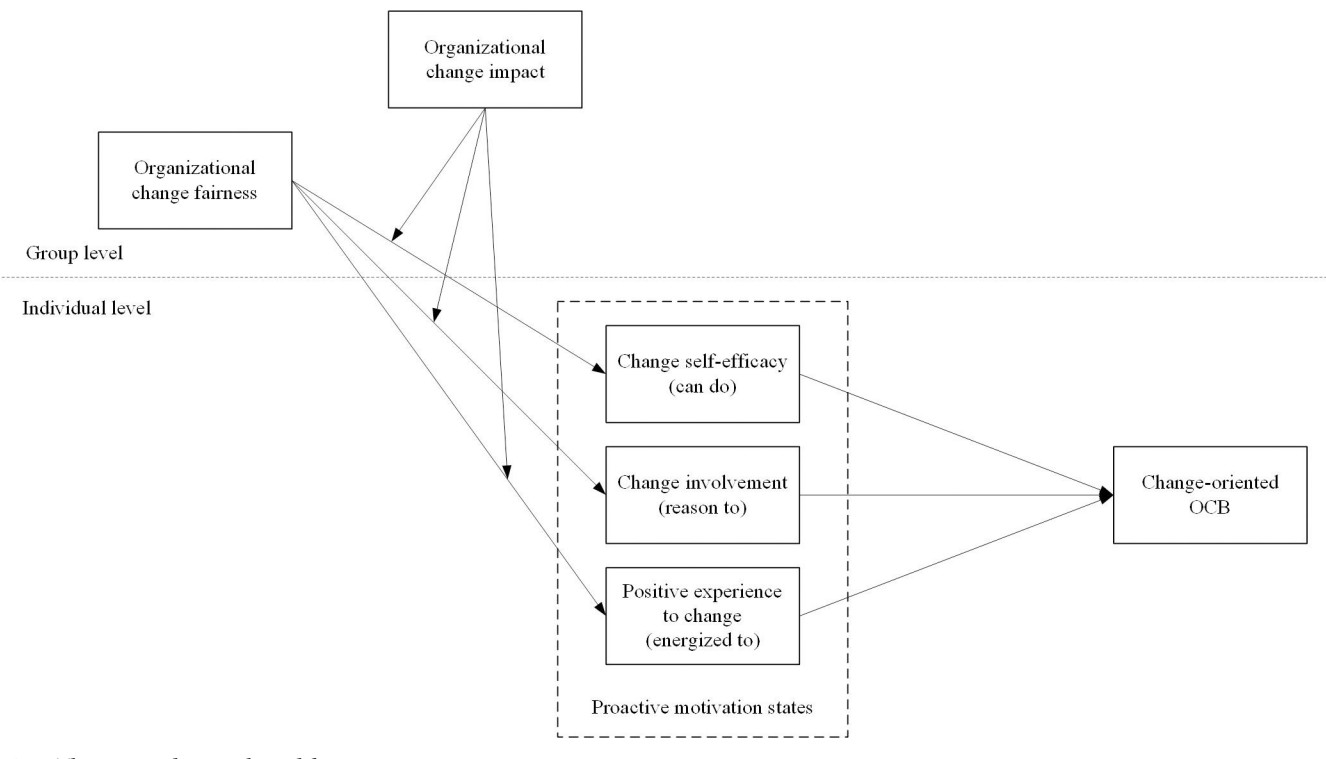

**Fig 1. The proposed research model.**

studies using a social exchange perspective show that fair treatment encourages reciprocation through extra-role behaviors [25]. Despite the established benefits of organizational fairness on OCB, few studies explore its impact on change-oriented OCB specifically. Hence, we propose Hypothesis 1.

**H1**: Group-level organizational change fairness will be positively related to change-oriented OCB.

## Group-level change fairness and change proactive motivation states

Understanding and uncovering employees' intrinsic motivation during the change process is crucial for enhancing the effectiveness of organizational change [13]. Traditional equity theory depicts employees as passive responders to their environment, while proactive motivation theory emphasizes employees' proactive role in shaping their surroundings [13]. For instance, employees have the capacity to self-set goals and self-motivate [26]. Proactive individuals align themselves with organizational objectives, demonstrate initiative, persist despite challenges, and drive change [27]. The level of personal initiative depends on the work environment [28]. According to Parker et al.'s research framework, contexts influence three proactive motivational states—"can do," "reason to," and "energized to"—thereby guiding individual proactivity [13]. We propose that organizational change fairness strengthens employees' proactive motivation.

First, self-efficacy, defined as the belief in one's ability to succeed, facilitates proactive goal-setting [29]. Individuals need confidence in their capabilities, known as "can do" motivation, to be motivated to set high goals, seek feedback, and actively engage [30]. "Can do" motivation reflects perceived abilities and outcome expectations [13], driving proactive activities such as

envisioning, planning, action, and reflection [31]. Our focus is on change self-efficacy, which encompasses perceptions of surpassing work tasks amidst organizational change initiatives. In highly proactive environments, greater confidence in achieving goals fosters stronger self-efficacy, predicting proactivity [32]. Correspondingly, research indicates that organizational change fairness encourages proactivity and enhances change self-efficacy [33].

Second, apart from confidence, individuals also require strong "reasons" to act proactively. We propose that intrinsic motivation and interest are significant drivers of proactive reasons. Intrinsic motivation often leads to self-initiated and persistent proactive behaviors, fostering positive and enduring changes. Proactivity often presents challenges and enjoyable experiences, fulfilling individuals' needs for competence and autonomy [13]. Simply put, desires for competence and autonomy inherently motivate proactive behaviors, like job involvement [34]. Our proposition is that by promoting fairness, change initiatives link employee involvement to intrinsic motivation. Those with higher intrinsic motivation exhibit a stronger internal desire to engage in change [35]. Supporting this, research reveals a positive association between organizational fairness and work involvement, with procedural and distributive fairness predicting involvement [36]. Moreover, ensuring employee involvement rights enhances perceived motivational fairness and refines processes [37]. These findings support our theory that organizational change fairness enhances change involvement.

The third aspect, apart from the "can do" and "reason to" states, is positive affect, a crucial pathway for being "energized to" engage in goal-setting and goal pursuit. Positive affect could aid employees in dealing with change by broadening their cognitive abilities and fostering flexibility [38], encouraging openness to solutions, and generating energy for new tasks [39]. Individuals experiencing positive affect tend to set more ambitious goals and invest more effort in achieving those goals. Previous studies have indicated that positive affect contributes to favorable change responses [35]. Employees, drawing from past experiences, use these experiences to comprehend subsequent changes [40]. We define positive change experience as general familiarity with comprehensive organizational changes [41]. Organizational change fairness may reinforce positive change experiences by promoting equitable, forward-thinking sustained behavior. Fairness implies respect for and confidence in employees' judgment, fostering positive affect [42]. Positive affect also directs attention toward favorable behavioral outcomes [43]. Employees, when treated respectfully, are more likely to accept change more readily [21]. Over time, accumulated experiences contribute to building change competence, readiness, and comfort [3].

In summary, organizational change fairness influences employee proactive motivation. Viewed through a proactive motivation framework, organizational change fairness seems to stimulate motivation by augmenting change self-efficacy, increasing work involvement, and fostering positive change experiences. Consequently, we propose the following:

**H2**: Group-level organizational change fairness will be positively related to individual-level proactive motivation states, including (a) change self-efficacy, (b) change involvement, and (c) positive experience to change.

## Group-level change impact as a moderator

The influence of change impact on individuals often receives inadequate attention in organizational change research. Herold [56]. define organizational change impact as the degree to which an organizational change affects an individual's job responsibilities and day-to-day work. Employees' change situation largely arises from new work unit demands and their influence on individual tasks [1]. Previous studies indicate that employees reassess their

organizational commitment based on evaluations of the change situation [4]. Notably, acceptance of change is partially contingent upon disruptions in daily work [44]. However, the understanding of change impact remains limited. While some may experience benefits from an organizational change, substantial evidence indicates that change frequently disrupts employees, resulting in increased workload, task conflicts, resource depletion, and concerns about failure, loss of control, insecurity, and uncertainty [44, 45]. In situations of significant change, heightened uncertainty may lead to reassessments of fairness [46]. When employees feel insecure or distressed, their motivation and readiness for change tend to diminish [39].

Studies suggest that greater change magnitudes lead to more negative outcomes and reactions [24, 47]. The impact of organizational change fairness on proactive motivational states varies with the change's magnitude. In low adaptation demand situations, fair procedures can satisfy proactive motivations and elicit positive reactions [48]. However, substantial impacts requiring extensive adjustment can diminish fairness' effectiveness and motivational influences, increasing employee risks and reducing confidence and readiness [24, 45]. Thus, we define organizational change impact at the group level, considering how changes affect a group's operations, dynamics, and functions. We propose Hypothesis 3:

**H3**: Group-level change impact negatively moderates the relationship between organizational change fairness and change proactive motivation states, including (a) change self-efficacy, (b) change involvement, and (c) positive experience to change. These relationships will become stronger in the low change impact condition rather than in the high change impact condition.

## Change proactive motivation states as a mediating mechanism

Up to this point, we've explored organizational factors like change fairness and impact on individuals' motivation for proactive change. Drawing from the framework of Parker, Bindl [13], proactive motivational states drive proactive behaviors, known to enhance job performance and outcomes such as OCB [39], highlighting their significance in organizational change.

First, extensive research has underscored the significant role of change self-efficacy in influencing organizational behaviors. This factor shapes individuals' perceptions of their ability to control change situations, instilling the confidence necessary to navigate through changes, impacting judgments concerning surpassing defined work duties, and forecasting responses to change such as acceptance, readiness, engagement, suggesting improvements, and commitment to change [39, 49].

Second, employee involvement in change is key to fostering change-oriented OCB, as engaged employees perceive their work as integral to their identity. Active participation in change initiatives is linked to positive change-related emotions, ongoing commitment, and enhanced work-related behaviors [50]. Involvement in change empowers employees by giving them a voice and choices, fostering support during change [51], and improving overall performance [52].

Third, positive change experiences significantly impact employee reactions, contributing to their understanding of change and triggering positive emotional responses, consequently correlating with favorable attitudes and behaviors towards change [45]. Employees lacking such experiences may struggle to comprehend the necessity for change, while those with substantial change management experience tend to adapt more effectively, understand the rationale behind change initiatives, stay updated, and proactively prepare for forthcoming transitions [53].

Drawing on Parker et al.'s perspective [13], which suggests that contextual factors predict proactivity distally and motivational states predict it proximally, our proposal posits that

organizational change fairness indirectly impacts change-oriented OCB by enhancing motivational states of "can do," "reason to," and "energized to." When organizational change fairness nurtures these motivational aspects, it leads to effective change-oriented OCB. Therefore, our hypothesis is as follows:

**H4**: Group-level organizational change fairness will have a mediating effect on change-oriented OCB through change proactive motivation states, including (a) change self-efficacy, (b) change involvement, and (c) positive experience to change.

# Method

## Participants and procedure

Using convenience sampling, we collected data from 107 work teams within 43 companies across various industries in Hangzhou, Zhejiang Province, China. After presenting our research objectives to a local industrial park's management, we randomly selected 60 companies. Of these, 43 companies, each having undergone at least one organizational change in the past year, agreed to participate and provided three teams each for the study. These companies were primarily from the finance, internet, logistics, telecommunications, cultural, and manufacturing sectors. We adhered to the ethical guidelines of the Declaration of Helsinki and received approval from the Ethics Committee of the corresponding author's university. Participation was voluntary, with informed consent obtained from all participants, and confidentiality maintained to avoid potential biases. Data collection, conducted from 01/04/2017 to 30/05/2017, utilized a two-source approach: team members reported on organizational change fairness, change impact, change self-efficacy, involvement, positive affect, and demographics; team leaders assessed change-oriented OCB. The survey was administered face-to-face using paper-based questionnaires.

We obtained usable data from 597 subordinates in 107 out of 129 teams across the 43 companies. On average, 5.58 participants ($SD$ = 2.38) completed the questionnaires per team, ranging from 2 to 13 members. The ratio of subordinates to teams and the sample size are similar to previous studies [54]. Regarding individual employee demographics, 39.4 percent of the participants were female. The average age and organizational tenure were 30.64 years ($SD$ = 5.94) and 4.46 years ($SD$ = 4.15), respectively. Furthermore, 53.8 percent of the participants received a 4-year undergraduate education, while 20.1 percent obtained a 3-year college education. Our data is accessible at https://osf.io/5xcun/?view_only= d29a262523c2444ab024d224f4e33579.

## Measurements

Participants rated the items on a scale from 1 = *strongly disagree* to 5 = *strongly agree*.

## Organizational change fairness

We used a three-item scale derived from Caldwell, Herold [55] to measure organizational change fairness. This scale was used to measure the perceived fairness characteristic of organizational change process. One sample item is "Sufficient advanced notice was given to employees affected by the change." The Cronbach's α was 0.86.

## Change impact

We used three items from the original six-time scale developed by Caldwell, Herold [55]. This scale was developed originally to measure individual job impact of organizational change. However, our study wants to capture the characteristics of the degree to which organizational

change affects employees' work process and procedures. The referent of the item was changed from *I/me/my* to *we/us/our* for the sake of constructing the team level of impact. One sample item is "The work processes and procedures we use have changed." The Cronbach's α was 0.79.

### Change self-efficacy

This variable was measured through a three-item scale from Herold, Fedor [56]. It reflects employees' change-specific belief in addressing demands and challenges raised by organizational change. One sample item is "I am able to successfully overcome the challenges of change." The Cronbach's α was 0.87.

### Change involvement

The three-item scale from previous study [57] was used to measure change involvement. This concept reflects the extent to which employees involve in organizational change. One sample item is "Supervisors discussed changes with me". The Cronbach's α was 0.77.

### Positive experience to change

We used a three-item scale to measure employees' positive experience to change. It reflects employees' affective states or attitude toward organizational change. This scale was established in previous study [53, 58]. One sample item is "I am excited about the change." The Cronbach's α was 0.85.

### Change-oriented OCB

A four-item scale derived from [6] was used to measure change-oriented OCB which reflects employees' constructive efforts to identify and implement organizational change programs. One sample item is "This employee frequently come up with new ideas or new work methods to perform their task". The Cronbach's α was 0.83.

### Control variables

We considered employee sex (male = 1 and female = 0), tenure (self-report), and education (high school and below = 1, three-year diploma = 2, undergraduate college = 3, postgraduate and above = 4) as control variables when running our data analysis.

### Confirmatory factor analysis and common method bias

We used Mplus 7.4 to test the confirmatory factor analysis and common method bias. Our hypothesized model included five variables: organizational change fairness, change impact, change self-efficacy, change involvement, and positive experience to change, along with change-oriented OCB. Confirmatory factor analysis of these variables demonstrated an excellent fit to our data (CFI = 0.97, TLI = 0.96, RMSEA = 0.05, SRMR = 0.04), outperforming alternative models as shown in Table 1.

To assess the potential impact of common method bias, we conducted Harman's single factor test. This test involved performing an exploratory factor analysis on all items used in the study. Using Principal Component Analysis (PCA) with SPSS, we extracted a single factor to examine the amount of variance it explained. The results indicated that the first factor accounted for 39% of the total variance, which is below the common threshold of 50% [59]. Therefore, it is unlikely that common method bias is a significant concern in our study. These

**Table 1. Results of confirmatory factor analysis.**

| Models | $\chi^2$/df | CFI | TLI | RMSEA | SRMR |
|---|---|---|---|---|---|
| Hypothesized model | 2.51 | .965 | .956 | .050 | .039 |
| Alternative model 1 | 5.84 | .884 | .860 | .090 | .071 |
| Alternative model 2 | 5.44 | .893 | .871 | .086 | .051 |
| Alternative model 3 | 5.96 | .881 | .856 | .091 | .078 |
| Alternative model 4 | 8.07 | .825 | .795 | .109 | .070 |
| Alternative model 5 | 11.23 | .742 | .704 | .131 | .096 |
| Alternative model 6 | 16.10 | .611 | .563 | .159 | .100 |

Notes: Alternative model 1 = combining change self-efficacy and job involvement into one factor; Alternative model 2 = combining change self-efficacy and positive experience to change; Alternative model 3 = combining change impact and change fairness; Alternative model 4 = combining three mediators; Alternative model 5 = combining change impact and change fairness into one factor and three mediators into another factor; Alternative model 6 = combining all indicators into one single factor.

findings suggest that our data is not substantially affected by common method bias, supporting the validity of our results.

## Aggregation test

Because we constructed organizational change fairness and change impact as the group-level concepts. They were aggregated from the individual-level values. Before aggregation, $r_{wg}$, $ICC_1$ and $ICC_2$ [60, 61] were computed to justify the aggregation test. The $r_{wg}$ score and ICC2 value of at least 0.70 indicate strong agreement among raters and sufficient reliability of group means, respectively. If ICC1 values exceed 0.05, it suggests a significant group effect or substantial between-group variance. The results indicated that the aggregation of change impact was supported with employee data ($ICC_1 = 0.12$, $ICC_2 = 0.43$, $r_{wg\_median} = 0.85$), while the aggregation of organizational change fairness also obtain a good support ($ICC_1 = 0.12$, $ICC_2 = 0.43$, $r_{wg\_median} = 0.82$). Although $ICC_2$ values were below 0.70, $ICC_1$ and $r_{wg}$ values still offered significant evidence in favor of data aggregation of these two variables.

## Analytical strategy

Given the multilevel nature of our model, we used hierarchical linear modeling (HLM, version 6.08) with full maximum likelihood estimation to explore cross-level relationships and multilevel mediation effects. We began with a null model test for the outcome variable, change-oriented OCB, which revealed significant between-team variance, $\chi^2(106) = 254.20$, $p < 0.001$, ICC1 = 0.20, indicating that 20% of the variance in change-oriented OCB occurs across teams. This finding underscores the necessity of multilevel analysis for our study.

To test the multilevel mediation effect, we used the procedure by Zhang, Zyphur [62], who recommend using a group-mean centering strategy for the individual-level mediator to increase the estimation precision for the mediating effects. Additionally, demographic variables such as gender, tenure, and education level were grand-mean centered to account for individual differences and enhance the robustness of the analysis. We have a 2-1-1 mediation model, in which a level-2 predictor influences a level-1 outcome through three level-1 mediators. Following the group-mean centering strategy, we centered all mediators by subtracting the group mean from the individual score and added their group means as level-2 predictors of change-oriented OCB. Thus, the between-group coefficient of the mediator rather than its within-group coefficient was used to estimate the size of our multilevel mediation effect. We used a Web application based on *RMediation* package (https://amplab.shinyapps.io/MEDCI/)

[63] to test the statistical significance and confidence interval (CI) of the multilevel mediation effects. For the cross-level moderating effects, we applied the grand-mean centering strategy for both level-2 predictors (i.e., organizational change fairness and change impact) as well as individual demographic variables.

## Result

### Descriptive statistics

Table 2 presents the means, standard deviations, and zero-order correlations for all variables. At the individual level, the outcome variable, change-oriented OCB, shows significant correlations with change self-efficacy ($r = 0.50$, $p < 0.01$), change involvement ($r = 0.43$, $p < 0.01$), and positive experience to change ($r = 0.50$, $p < 0.01$). Change self-efficacy is correlated with change involvement ($r = 0.37$, $p < 0.01$) and positive experience to change ($r = 0.59$, $p < 0.01$), and change involvement also correlates with positive experience to change ($r = 0.44$, $p < 0.01$). These variables, aggregated to the group level, continue to show significant inter-correlations.

At the group level, organizational change fairness correlates with change impact ($r = 0.51$, $p < 0.001$), and both variables show significant correlations with group-aggregated outcome variables: organizational change fairness is linked to change self-efficacy ($r = 0.32$, $p < 0.01$), change involvement ($r = 0.34$, $p < 0.01$), and positive experience to change ($r = 0.49$, $p < 0.01$). Change impact also correlates with these outcomes: change self-efficacy ($r = 0.38$, $p < 0.01$), change involvement ($r = 0.36$, $p < 0.01$), and positive experience to change ($r = 0.66$, $p < 0.01$).

### Hypotheses tests

Multilevel hypotheses tested with HLM are shown in Table 3. Hypothesis 1 expects a positive impact of group-level organizational change fairness on change-oriented OCB, and Model 1 supports this hypothesis ($\gamma = 0.50$, $p < 0.001$).

**Table 2. Results of descriptive statistics, correlations and aggregation test.**

| Variable | M | SD | ICC1 | ICC2 | $r_{wg}$ | 1 | 2 | 3 | 4 | 5 | 6 | 7 | 8 | 9 | 10 | 11 | 12 |
|---|---|---|---|---|---|---|---|---|---|---|---|---|---|---|---|---|---|
| *Individual level* | | | | | | | | | | | | | | | | | |
| 1. Gender | 0.39 | 0.49 | | | | – | .09* | .04 | -.08 | -.03 | -.08* | -.06 | | | | | |
| 2. Education | 2.72 | 0.85 | | | | | – | -.12** | -.02 | .03 | .01 | -.04 | | | | | |
| 3.Tenure | 4.46 | 4.15 | | | | | | – | -.05 | .10* | -.05 | .07 | | | | | |
| 4. CSE | 4.01 | 0.67 | .18 | .54 | .90 | | | | **.87** | .37** | .59** | .50** | | | | | |
| 5. CIV | 3.67 | 0.78 | .14 | .48 | .74 | | | | | **.77** | .44** | .43** | | | | | |
| 6. PEC | 3.71 | 0.76 | .22 | .61 | .85 | | | | | | **.85** | .50** | | | | | |
| 7. Change-oriented OCB | 3.85 | 0.69 | .20 | .58 | .78 | | | | | | | **.83** | | | | | |
| *Group level* | | | | | | | | | | | | | | | | | |
| 8. CSE Mean | 3.99 | 0.37 | | | | | | | | | | | – | | | | |
| 9. CIV mean | 3.64 | 0.45 | | | | | | | | | | | .40** | – | | | |
| 10. PEC mean | 3.67 | 0.45 | | | | | | | | | | | .67** | .50** | – | | |
| 11. Change impact mean | 3.67 | 0.40 | .12 | .43 | .85 | | | | | | | | .32** | .34** | .49** | **.79** | |
| 12. Change fairness mean | 3.60 | 0.40 | .12 | .43 | .82 | | | | | | | | .38** | .36** | .66** | .51** | **.86** |

Note: CSE = change self-efficacy, CIV = change involvement, PEC = positive experience to change. Correlation coefficients above the diagonal are analyses at the individual level (N = 597). Correlation coefficients below the diagonal are analyses at the team level (N = 107). $R_{wg}$ is quantified by median

*$p < .05$

**$p < .01$

***$p < .001$.

**Table 3. Results of hierarchical linear modeling.**

| | Null model | Total effect | Change self-efficacy | | | Change involvement | | | Positive experience to change | | |
|---|---|---|---|---|---|---|---|---|---|---|---|
| **Dependent variables** | OCB | OCB | CSE | OCB | CSE | CI | OCB | CI | PEC | OCB | PEC |
| **Model** | M0 | M1 | M2 | M3 | M4 | M5 | M6 | M7 | M8 | M9 | M10 |
| *Individual level* | | | | | | | | | | | |
| Intercept | 3.83*** | 3.83*** | 3.99*** | 3.83*** | 4.04*** | 3.65*** | 3.83*** | 3.67*** | 3.68*** | 3.82*** | 3.72*** |
| Gender | | -0.09 | -0.10** | -0.04 | -0.09 | -0.08 | -0.06 | -0.07 | -0.16** | -0.03 | -0.15** |
| Education | | -0.003 | -0.01 | 0.00 | -0.02 | 0.04 | -0.01 | 0.03 | 0.01 | -0.01 | -0.004 |
| Tenure | | 0.01 | -0.01 | 0.01** | -0.01 | 0.02** | 0.004 | 0.02** | -0.01 | 0.01* | -0.01 |
| CSE | | | | 0.44*** | | | | | | | |
| CIV | | | | | | | 0.34*** | | | | |
| PEC | | | | | | | | | | 0.38*** | |
| *Group level* | | | | | | | | | | | |
| Change impact | | | | | 0.15 | | | 0.20 | | | 0.17 |
| Change fairness | | *0.50****\ | **0.36**** | *0.30*** | *0.31*** | **0.38***** | *0.37*** | *0.30** | **0.74***** | *0.10* | *0.69***** |
| Interaction | | | | | -0.53† | | | -0.28 | | | -0.47† |
| CSE Mean | | | | **0.54***** | | | | | | | |
| CIV Mean | | | | | | | **0.31***** | | | | |
| PEC Mean | | | | | | | | | | **0.54***** | |
| $\sigma^2$ | 0.375 | 0.374 | 0.361 | 0.302 | 0.362 | 0.512 | 0.316 | 0.513 | 0.444 | 0.310 | 0.443 |
| $\tau$ | 0.095 | 0.056 | 0.060 | 0.036 | 0.052 | 0.066 | 0.050 | 0.061 | 0.043 | 0.034 | 0.038 |
| $R^2_{within-group}$ | | 0.002 | 0.01 | 0.19 | 0.003 | 0.01 | 0.16 | 0.01 | 0.02 | 0.17 | 0.02 |
| $R^2_{between-group}$ | | 0.41 | 0.23 | 0.38 | 0.33 | 0.23 | 0.11 | 0.29 | 0.65 | 0.39 | 0.69 |

Notes

$^\dagger p < .10$

$^* p < .05$

$^{**} p < .01$

$^{***} p < .001$. $N_{employee}$ = 597, $N_{team}$ = 107. CSE = change self-efficacy, CIV = change involvement, PEC = positive experience to change. Interaction = change fairness × change impact.

Furthermore, we assumed that organizational change fairness is related to individual-level change proactive motivation states. As results shown in Table 3, group-level organizational change fairness had significant effects on change self-efficacy ($\gamma$ = 0.36, $p$ < 0.01), change involvement ($\gamma$ = 0.38, $p$ < 0.001), positive experience to change ($\gamma$ = 0.74, $p$ < 0.001) at individual level. Hence, hypotheses 2(H2a-H2c) were supported.

Hypothesis 3 predicts that change impact negatively moderates the relationship between group-level organizational change fairness and change proactive motivational states. Our results indicate that there is a marginally significant cross-level relationship between group-level change impact and change fairness interactions for change self-efficacy ($\gamma$ = -0.53, $p$ = 0.067) and positive experience to change ($\gamma$ = -0.47, $p$ = 0.054), as shown in Model 4 and Model 10 in Table 3. However, the interaction between change impact and change fairness at group level was not significant for change involvement ($\gamma$ = -0.28, $p$ = 0.427), as shown in Model 7. The results indicate that as the group-level change impact increases, there is a significant decrease in the influence of group-level organizational change fairness on change self-efficacy and positive experience.

To investigate the interaction further, we plot the simple slope effect of group-level change impact at high (+1 $SD$) and low (1 $SD$) levels. Fig 2 shows that the relationship between group-level organizational change fairness and change self-efficacy turns from a positive to a negative

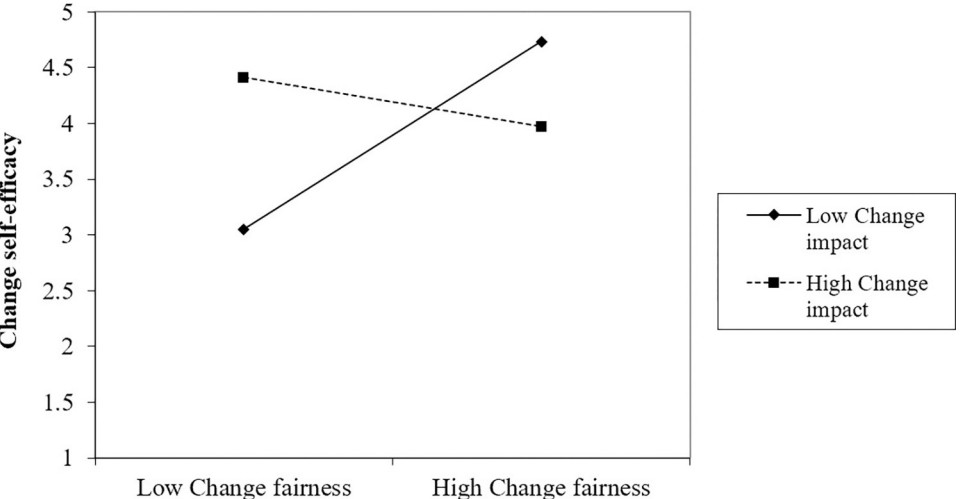

**Fig 2. Change impact moderates the effect of organizational change fairness on change self-efficacy.**

correlation as change impact increases compared to low levels of change impact. This indicates that group-level organizational change fairness has a weakened influence on employees' change self-efficacy in those teams with more change impact. Similarly, Fig 3 shows that group-level organizational change fairness has a greater positive relationship with employees' positive experience to change when change impact was smaller rather than larger. Thus, Hypothesis 3 is partially supported (H3a, H3c are supported, H3b is not supported).

The results suggest that group-level organizational change fairness significantly influences change self-efficacy, change involvement, and positive experience to change, confirming the path from the independent variable to the mediators. Subsequent analysis will estimate path coefficients from these mediators to the outcome variable, change-oriented OCB, and assess direct effects of organizational change fairness while controlling for demographic factors.

Table 3 shows that change self-efficacy ($\gamma = 0.44$, $p < 0.001$), change involvement ($\gamma = 0.34$, $p < 0.001$), and positive experience to change ($\gamma = 0.38$, $p < 0.001$) significantly enhance

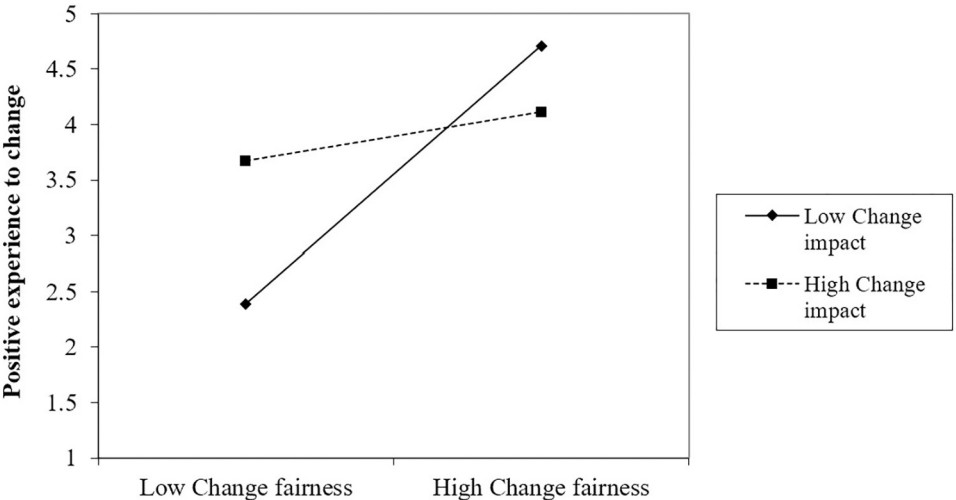

**Fig 3. Change impact moderates the effect of change fairness on positive experience to change.**

**Table 4. Results for multilevel mediation effects by *RMediation*.**

| Mediators | Path *a* | | Path *b* | | Indirect effects | 95%CI |
|---|---|---|---|---|---|---|
| | estimate | SE | estimate | SE | | |
| Change self-efficacy | 0.36[**] | 0.09 | .54[***] | .14 | .19(.07) | [0.071, 0.35] |
| Change involvement | 0.38[***] | 0.12 | .31[***] | .08 | .12(.05) | [0.036, 0.226] |
| Positive experience to change | .74[***] | .08 | .54[***] | .11 | .40 (.09) | [0.229, 0.591] |

change-oriented OCB. Hypothesis 4 posits that these factors mediate the relationship between organizational change fairness and change-oriented OCB. When group-level organizational change fairness was used as the independent variable and control variables were included, the positive effects of change self-efficacy, change involvement, and positive experience to change on change-oriented OCB were supported (Model 3: $\gamma = 0.54$, $p < 0.001$; Model 6: $\gamma = 0.31$, $p < 0.001$; Model 9: $\gamma = 0.54$, $p < 0.001$). Meanwhile, the positive effect of change fairness on change-oriented OCB decreased from 0.50 ($p < 0.001$) to 0.30 ($p < 0.01$), 0.37 ($p < 0.01$), 0.10 ($p > .05$), respectively. Thus, the mediation effects were evident as the positive impact of change fairness on change-oriented OCB remained significant across various models, with a reduction in the effect of change impact over successive models.

Moreover, we used the Monte Carlo method to estimate the confidence intervals (CIs, 95% significant level). The results from Table 4 indicated that change self-efficacy ($ab = 0.19$, $ab_{se} = 0.07$, 95% CIs [0.07, 0.35]), change involvement ($ab = 0.12$, $ab_{se} = 0.05$, 95% CIs [0.04, 0.23]) and positive experience to change ($ab = 0.40$, $ab_{se} = 0.09$, 95% CIs [0.23, 0.59]) would play significant mediating roles between change impact and change-oriented OCB. These results support Hypothesis 4 (H4a, H4b, and H4c), indicating significant mediating roles for these variables between organizational change fairness and change-oriented OCB, with effect sizes confirming the strength of these mediations.

## Discussion

This study used multilevel analysis to explore the influence of contextual factors, namely organizational change fairness and change impact, on employees' proactive motivation and behaviors in relation to organizational change. The study reveals that team-level organizational change fairness significantly correlated with change-oriented OCB and three key proactive motivational states: change self-efficacy, involvement, and positive change experience. Moreover, this study highlights that team-level change impact moderated the relationships between fairness and self-efficacy, as well as positive experience, but not involvement. In instances of low change impact, the favorable effects of fairness on self-efficacy and experience were intensified. Conversely, these effects diminished under conditions of high change impact. Furthermore, our findings indicate that the motivational states served as mediators in the fairness-OCB relationship. Prior studies have stated that OCB plays a key role in the implementation of change, but relevant research mostly focused on the stable environment and less attention is paid to the mechanism affecting change-oriented OCB in the context of change. This study draws upon the proactive motivation model proposed by Parker et al. and the existing literature on organizational change context [13, 44], to extend the understanding of how these factors shape employees' attitudes and actions during change initiatives. This study emphasized the important influence of organizational change fairness on change-oriented OCB, and the influence of active motivation state on this behavior. The results of this study conveyed the importance of fairness in the process of change to organizational managers, strengthened their

understanding of employees' attitude and behavior, and provided effective practical suggestions for promoting and implementing organizational change.

## Theoretical implication

This study introduces a multilevel framework to explore the antecedents of change-oriented OCB within organizational change contexts. While the literature has extensively examined both contextual [6, 7] and individual determinants [12, 64] of change-oriented OCB, less research has focused on the contextual mechanisms driving this behavior. Unlike traditional forms of OCB [65, 66], change-oriented OCB represents proactive behavior that facilitates changes in work processes and outcomes [64], making it crucial for organizational transitions. Our findings highlight the significant influence of group-level organizational change fairness, positioning it as a key factor that promotes engagement in change-oriented OCB during implementation phases.

Second, this study significantly deepens the theoretical understanding of the relationship between organizational fairness and change-oriented OCB. It extends the proactive motivation model (Parker et al., 2010) by linking group-level organizational change fairness to three proactive states underpinning change-oriented OCB. Drawing from organizational change literature [56], we conceptualize organizational change fairness as the collective perception of fairness among employees during the implementation phase. This shared perspective facilitates active engagement in change-related activities. Cultivating a mutual sense of fairness holds paramount importance for organizations aiming to encourage active participation, bolster self-efficacy, and promote positive change experiences. However, the influence of fairness varies, with change impact acting as a moderating factor that can either amplify or diminish its effects. This suggests that the scale of change impact can either hinder or mitigate the positive effects of fairness. The implementation of change often disrupts workflows, increases workloads, and alters tasks [48], thus creating new demands. Group-level organizational change fairness and its impact together form a significant change context that influences proactive motivational states. Our findings substantiate the role of contextual variables as pivotal predictors of motivation, aligning with the insights put forth by Parker et al. (2010). In essence, the integration of fairness and impact contributes to a deeper understanding of antecedents related to proactive states and their impact in the change process.

Third, our study introduces a novel perspective by exploring proactive motivational states as mediating mechanisms, thereby advancing both the proactive motivation model and research on change-oriented OCB. Past research [16] has shown that proactive motivation is linked to the emergence of proactive behaviors. However, our study differs by embedding this motivational model within organizational change contexts, aiming to shed light on the mechanisms that affect change-oriented OCB. We specifically identify change-specific motivational factors—change self-efficacy, involvement, and positive change experience—as mediators. Our empirical findings robustly support the significance of these three mediators, aligning with arguments proposed by Parker et al. (2010), which highlight the pursuit of diverse proactive goals crucial for implementing organizational change. Consequently, this study contributes empirical evidence by contextualizing the proactive motivational model within change environments, elucidating the influence processes on change-oriented OCB.

## Practical implication

This study offers actionable implications for organizational managers. First, recognizing the importance of change-oriented OCB is crucial, especially during the implementation phase to mitigate reactive resistance to change. Managers should actively promote this type of behavior

by reinforcing employees' identification with and commitment to the change through enhanced communication and education. Specific actions include conducting regular workshops and training sessions that focus on the benefits and goals of the change, thus aligning employees' personal and professional objectives with the organizational change mission. Additionally, establishing incentive mechanisms to reward change-oriented behaviors can further boost employees' enthusiasm and creativity, playing a vital role in the change process.

Second, our findings highlight the critical role of perceived organizational change fairness in fostering proactive behavior. Managers should prioritize improving employees' fairness perceptions to facilitate smooth change implementation. This can be achieved by involving employees in the development of change plans, ensuring transparency in the processes, and communicating regularly about the progress and benefits of change initiatives. Practical steps include creating inclusive committees or focus groups that allow employee input on change strategies, and maintaining open channels of communication to address concerns and provide updates. Such involvement helps eliminate doubts and misunderstandings, thereby enhancing support for the change.

Third, organizations should cultivate proactive motivational states such as "can do," "reason to," and "energized to," which positively influence change-oriented OCB. The unpredictable nature of change requires proactive mindsets. Managers can support this by providing targeted training and development programs that support to enhance skills and build confidence. These programs should include stress management workshops, resilience training, and opportunities for employees to practice new skills in a supportive environment. Additionally, attending to employees' psychological and emotional reactions, and implementing measures to alleviate stress and anxiety. Recognizing and rewarding employees as they achieve milestones in the change process not only boosts morale but also reinforces the value of their contributions to successful outcomes.

## Future direction and limitation

This study has certain limitations that future research should address. Firstly, while we operationalized proactive motivational states as change self-efficacy, involvement, and positive experience, alternative conceptualizations of these states may offer deeper insights, as suggested by Parker et al. (2010). Secondly, our cross-sectional design limits causal inference; longitudinal studies are recommended to better understand the dynamics of change-oriented OCB over time. Lastly, further exploration of additional contextual factors like change continuity, transparency, and disruptiveness is needed beyond group-level organizational change fairness and impact [67].

## Author Contributions

**Conceptualization:** Bin Ling, Dusheng Chen.

**Data curation:** Qu Yao.

**Formal analysis:** Bin Ling, Qu Yao, Yutong Liu.

**Investigation:** Bin Ling.

**Methodology:** Qu Yao, Yutong Liu.

**Resources:** Dusheng Chen.

**Supervision:** Bin Ling.

**Writing – original draft:** Bin Ling, Qu Yao, Yutong Liu.

**Writing – review & editing:** Bin Ling, Qu Yao, Yutong Liu, Dusheng Chen.

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
