## [Decision Letter · Decision Letter 0]

12 Jul 2024

PONE-D-24-21250Fairness Matters for Change: A Multilevel Study on Organizational Change Fairness, Proactive Motivation, and Change-Oriented OCBPLOS ONE

Dear Dr. Ling,

Thank you for submitting your manuscript to PLOS ONE. After careful consideration, we feel that it has merit but does not fully meet PLOS ONE’s publication criteria as it currently stands. Therefore, we invite you to submit a revised version of the manuscript that addresses the points raised during the review process.

**This manuscript has many advantages. But, I think the authors need to add additional tables.**

We look forward to receiving your revised manuscript.

Kind regards,

Chunyu Zhang

Academic Editor

PLOS ONE

Journal Requirements:

This research was supported by Soft Science Research Plan Project for Nanjing (202303001)

3. In the online submission form, you indicated that The datasets analyzed during the current study are available from the corresponding author on reasonable request.

5. We note you have included a table to which you do not refer in the text of your manuscript. Please ensure that you refer to Tables 1, 2, 3 and 4 in your text; if accepted, production will need this reference to link the reader to the Table.

6. Please ensure that you include a title page within your main document. We do appreciate that you have a title page document uploaded as a separate file, however, as per our author guidelines (http://journals.plos.org/plosone/s/submission-guidelines#loc-title-page) we do require this to be part of the manuscript file itself and not uploaded separately.

Reviewers' comments:

Reviewer's Responses to Questions

**Comments to the Author**

1. Is the manuscript technically sound, and do the data support the conclusions?

Reviewer #1: Yes

Reviewer #2: Yes

2. Has the statistical analysis been performed appropriately and rigorously? 

Reviewer #1: Yes

Reviewer #2: Yes

3. Have the authors made all data underlying the findings in their manuscript fully available?

Reviewer #1: No

Reviewer #2: No

4. Is the manuscript presented in an intelligible fashion and written in standard English?

Reviewer #1: Yes

Reviewer #2: Yes

5. Review Comments to the Author

**Reviewer #1:** • The abstract needs to be expanded upon and improved. Crucial information on the origin, methodology, conclusion, and implication needs to be included.

• In the section of introduction and review, highlight the significance of studying the relationship between fairness, motivation, and OCB in the context of organizational change. The authors need to clearly address the gap as to why this study is essential.

• Result tables are not present in the PDF file of the manuscript.

Broadly, the manuscript is well-structured, and academically rigorous, providing valuable insights into the role of fairness in organizational change and its effects on proactive motivation and change-oriented OCB.

**Reviewer #2:** I am very grateful for the opportunity to review this paper on organizational change fairness and change-oriented OCB. The research questions in this paper are very interesting, and I greatly appreciate the theoretical mechanisms of proactive motivation presented in it.

1.The introduction is well-written, but the author needs to more clearly explain why it is necessary to use the proactive motivation framework to interpret the relationship between organizational change fairness and change-oriented OCB. This would help in better understanding the research motivation of this paper.

2.The literature in this paper is outdated; I recommend that the authors supplement it with newer sources.

3.On page ten, "employee involvement" should be highlighted as involvement in change. It is recommended to emphasize in the writing that it refers to employees' involvement in change, not just general employee work involvement.

4.The author uses both single and double quotation marks interchangeably, such as 'can do,' 'reason to,' and 'energized to.' It is recommended to use double quotation marks consistently throughout the text.

5.The paper lacks an analysis of common method bias; it is recommended that the author supplement this analysis.

6.In the discussion section, the author is advised to deepen the theoretical significance of the relationship between organizational fairness and change OCB. In the practical implications section, enhance the discussion on the significance of the research findings for practical activities, and link them to specific organizational change activities such as change implementation management and change training.

7.The paper lacks the presentation of four results tables; it is recommended to include results from Table 1 to Table 4.

6. PLOS authors have the option to publish the peer review history of their article (what does this mean?). If published, this will include your full peer review and any attached files.

Reviewer #1: **Yes: **Dr. Abhishek Sharma

Reviewer #2: No

---

## [Author Response · Author response to Decision Letter 0]

14 Aug 2024

Q1: The abstract needs to be expanded upon and improved. Crucial information on the origin, methodology, conclusion, and implication needs to be included. 

Response: 

Thank you for your valuable feedback on our manuscript. We have made the following revisions in response to your suggestion to expand and improve the abstract:

Origin: At the beginning of the abstract, we have clarified the background and motivation of the study, highlighting the importance of achieving consensus on fairness within the team for the success of organizational change.

Methodology: We have detailed the theoretical foundation of the study, specifically the proactive motivation model, and the establishment of a multilevel framework. Additionally, we have mentioned the data source, which includes 597 employees from 107 teams across 43 Chinese companies.

Conclusion: The abstract now clearly presents the main findings, stating that group-level perceived organizational change fairness significantly predicts employees' change-oriented OCB through organizational change self-efficacy, involvement, and positive emotional experiences.

Implication: We have further elaborated on the practical applications and theoretical implications of our findings, emphasizing the importance of these insights for understanding the cross-level determinants influencing change-oriented OCB.

Thank you again for your suggestions, which have helped us to improve the quality of our manuscript. We appreciate any further feedback you may have. 

Changes: Page 1

Q2: In the section of introduction and review, highlight the significance of studying the relationship between fairness, motivation, and OCB in the context of organizational change. The authors need to clearly address the gap as to why this study is essential.

Response: 

Thank you for your valuable feedback on our study. We have made the following revisions in response to your suggestion to highlight the significance of studying the relationship between fairness, motivation, and OCB in the context of organizational change in the introduction and literature review sections:

1. In the Introduction Section:

Emphasizing the Study’s Importance. We added descriptions of the critical role of organizational change in modern business practices, particularly how perceptions of fairness impact the success of change (Arnéguy et al., 2018; Khaw et al., 2023; Xu et al., 2016). We emphasized the importance of leaders demonstrating procedural fairness during the change process to enhance employees' acceptance of change (Kim et al., 2023).

Clarifying the Study’s Significance. We elaborated on the importance of understanding how team-level perceptions of organizational change fairness influence employees’ proactive motivational states to promote change-oriented OCB. This perspective contributes to theoretical development and provides practical guidance for management practices.

2. In the Literature Review Section:

Identifying the Research Gap. We highlighted the shortcomings of current literature in exploring the relationship between fairness, motivation, and OCB in the context of organizational change. While existing studies have examined these variables individually, few have investigated their interactions and manifestations within a multilevel framework (Choi, 2007; Li et al., 2016; Seppälä et al., 2012).

Detailing the Study’s Importance. In the literature review, we specifically discussed how fairness influences proactive motivational states, which in turn promote change-oriented OCB. We referenced Parker et al. (2010)’s proactive motivation model to explain how the three motivational states—“can do,” “reason to,” and “energized to”—drive individual proactive behaviors (Berg & Kauffeld, 2024; Frese & Fay, 2001). We also discussed how perceptions of fairness stimulate employees’ proactive tendencies, fostering change-oriented OCB by enhancing change self-efficacy, involvement, and positive emotional experiences. Furthermore, we pointed out that these proactive motivational states might be amplified or diminished by the moderating effect of change impact.

We believe these changes will improve the clarity and coherence of our research.

Changes: Page From 2 to 11

Q3: Result tables are not present in the PDF file of the manuscript.

Response: Thank you for pointing out the issue with our manuscript. We have corrected this oversight and have submitted a revised version of the manuscript that includes all the necessary result tables and figures. The specific modifications are as follows:

1. Table 1: Results of Confirmatory Factor Analysis:

Includes �2/df, CFI, TLI, RMSEA, and SRMR values for the hypothesized model and several alternative models.

2. Table 2: Results of Descriptive Statistics, Correlations, and Aggregation Test:

Provides means (M), standard deviations (SD), ICC1, ICC2, and rwg values for each variable, as well as correlation coefficients at both the individual and team levels.

3. Table 3: Results of Hierarchical Linear Modeling:

Shows the results of the null model, total effect model, and hierarchical analysis of change self-efficacy, change involvement, and positive experience to change.

4. Table 4: Results for Multilevel Mediation Effects by RMediation:

Provides mediation effect results, including path a estimate, path a standard errors, path b estimates, path b standard errors, indirect effects, and 95% confidence intervals.

These tables are now fully included in the revised manuscript to better support our research findings. We appreciate your attention to this matter and look forward to your further review.

Changes: Page From 14, 18, 19, 21 to 22

Response to Reviewer 2:

Q1: I am very grateful for the opportunity to review this paper on organizational change fairness and change-oriented OCB. The research questions in this paper are very interesting, and I greatly appreciate the theoretical mechanisms of proactive motivation presented in it.

1.The introduction is well-written, but the author needs to more clearly explain why it is necessary to use the proactive motivation framework to interpret the relationship between organizational change fairness and change-oriented OCB. This would help in better understanding the research motivation of this paper.

Response: 

Thank you for your positive feedback and valuable suggestions on our manuscript. In response to your suggestion to more clearly explain why it is necessary to use the proactive motivation framework to interpret the relationship between organizational change fairness and change-oriented OCB in the introduction, In the Introduction Section we have made the following revisions:

1. Necessity of Using the Proactive Motivation Framework 

We have added a paragraph that explains in detail how the proactive motivation framework systematically elucidates the influence of individual motivational states (“can do,” “reason to,” and “energized to”) on proactive behaviors. These motivational states help better explain why employees, when perceiving fairness in the organizational change process, exhibit higher levels of change-oriented OCB. Specifically, perceived fairness can stimulate employees’ change self-efficacy, involvement, and positive emotional experiences, thus encouraging them to proactively support organizational change.

2. Theoretical Support

We have cited Parker et al. (2010) to further illustrate how the proactive motivation framework aids in understanding individuals’ motivation and behavior in the face of change. We also discussed how existing research supports the use of this theoretical framework in explaining the relationship between fairness in organizational change and change-oriented OCB.

We believe these revisions will better clarify the research motivation and theoretical foundation of our study. We appreciate any further feedback. 

Changes: Throughout the manuscript.

Q2: The literature in this paper is outdated; I recommend that the authors supplement it with newer sources.

Response: Thank you for pointing out the importance of precision in our choice of terminology. Thank you for your valuable feedback on our manuscript. In response to your comment regarding the outdated literature, we have thoroughly updated our references to include more recent studies, ensuring the content is both current and relevant. The specific changes are as follows:

We have added the following references: Berg and Kauffeld (2024); Fullerton et al. (2021); Kalra et al. (2021); Khaw et al. (2023); Liao (2023); Si et al. (2023); Unterhitzenberger and Lawrence (2023). These references provide the latest research findings and theoretical support, enhancing the foundation of our study.

Simultaneously, we have removed the following outdated references: Bateman and Crant (1993); Brotheridge (2003); Glomb et al. (2011); Herold et al. (2008); Riolli and Savicki (2006); Roberson (2006); Sagie et al. (1990); Tyler and De Cremer (2005). These references have been replaced by more recent and relevant studies.

With these updates, we have ensured that the references cited are up-to-date and pertinent. We believe these changes will significantly enhance the academic contribution and practical relevance of our paper. Thank you again for your valuable suggestions. 

Changes: Throughout the manuscript.

Q3: On page ten, “employee involvement” should be highlighted as involvement in change. It is recommended to emphasize in the writing that it refers to employees’ involvement in change, not just general employee work involvement.

Response: Thank you for your valuable suggestion. We have clarified “employee involvement” as “involvement in change” on page ten and emphasized that it specifically refers to employees’ involvement in change, not just general work involvement.

Changes: Throughout the manuscript.

Q4: The author uses both single and double quotation marks interchangeably, such as 'can do,' 'reason to,' and 'energized to.' It is recommended to use double quotation marks consistently throughout the text.

Response: Thank you for your feedback. We have consistently used double quotation marks throughout the text to ensure uniformity.

Changes: Throughout the manuscript.

Q5: The paper lacks an analysis of common method bias; it is recommended that the author supplement this analysis.

Response: Thank you for your suggestion. We have supplemented the analysis of common method bias (CMB) using Harman’s Single Factor Test and reported the results in the main text.

Changes: Page 5 and page from 12 to 14

Q6: In the discussion section, the author is advised to deepen the theoretical significance of the relationship between organizational fairness and change OCB. In the practical implications section, enhance the discussion on the significance of the research findings for practical activities, and link them to specific organizational change activities such as change implementation management and change training.

Response: Thank you for your valuable feedback on our discussion and practical implications sections. We have made the following revisions based on your suggestions:

In the theoretical implications section, we have deepened the understanding of the relationship between organizational fairness and change-oriented OCB. We extended Parker et al. (2010)'s proactive motivation model by linking group-level organizational change fairness to three proactive states that support change-oriented OCB, emphasizing the importance of organizational change fairness in promoting active engagement in change activities. Additionally, we explored the moderating role of change impact on the perception of fairness, clarifying the interaction between change impact and fairness perception.

In the practical implications section, we enhanced the discussion on the significance of our findings for practical activities and linked them to specific organizational change activities such as change implementation management and change training. We recommend that managers enhance employee identification and commitment to change through regular workshops and training sessions and establish incentive mechanisms to encourage change-oriented behaviors. Furthermore, we highlighted the importance of improving employees' fairness perceptions through transparent communication and employee involvement to support successful organizational change implementation.

Changes: Page from 22 to 25

Q7: The paper lacks the presentation of four results tables; it is recommended to include results from Table 1 to Table 4.

Response: Thank you for your valuable suggestion. We have included the results from Table 1 to Table 4 in the results section of the paper as per your recommendation. For detailed changes, please refer to Reviewer 1, Q3 (pp. 3-4 in this letter). We believe this modification improves the organization of the manuscript and effectively addresses your concerns.

Changes: Page From 14, 18, 19, 21 to 22

Reference

Arnéguy, E., Ohana, M., & Stinglhamber, F. (2018). Organizational justice and readiness for change: A concomitant examination of the mediating role of perceived organizational support and identification. Frontiers in Psychology, 9, 1-13. https://doi.org/10.3389/fpsyg.2018.01172

Bateman, T. S., & Crant, J. M. (1993). The proactive component of organizational behavior: A measure and correlates. Journal of Organizational Behavior, 14, 103-118. https://doi.org/10.1002/job.4030140202

Berg, A.-K., & Kauffeld, S. (2024). Proactive verbal behavior in team meetings: Effects of supportive and critical responses on satisfaction and performance. Current Psychology, 43(23), 20640-20654. https://doi.org/10.1007/s12144-024-05806-y

Brotheridge, C. M. (2003). The role of fairness in mediating the effects of voice and justification on stress and other outcomes in a climate of organizational change. International Journal of Stress Management, 10(3), 253-268. https://doi.org/10.1037/1072-5245.10.3.253

Choi, J. N. (2007). Change-oriented organizational citizenship behavior: Effects of work environment characteristics and intervening psychological processes. Journal of Organizational Behavior, 28(4), 467-484. https://doi.org/https://doi.org/10.1002/job.433

Frese, M., & Fay, D. (2001). Personal initiative: An active performance concept for work in the 21st century. Research in Organizational Behavior, 23, 133-187. https://doi.org/10.1016/S0191-3085(01)23005-6 

Fullerton, D. J., Zhang, L. M., & Kleitman, S. (2021). An integrative process model of resilience in an academic context: Resilience resources, coping strategies, and positive adaptation. PLOS ONE, 16(2), e0246000. https://doi.org/10.1371/journal.pone.0246000

Glomb, T. M., Bhave, D. P., Miner, A. G., & Wall, M. (2011). Doing good, feeling good: Examining the role of organizational citizenship behaviors in changing mood [Article]. Personnel Psychology, 64(1), 191-223. https://doi.org/10.1111/j.1744-6570.2010.01206.x

Herold, D. M., Fedor, D. B., Caldwell, S., & Liu, Y. (2008). The effects of transformational and change leadership on employees' commitment to a change: A multilevel study [Article]. Journal of Applied Psychology, 93(2), 346-357. https://doi.org/10.1037/0021-9010.93.2.346

Kalra, A., Agnihotri, R., Singh, R., Puri, S., & Kumar, N. (2021). Assessing the drivers and outcomes of behavioral self-leadership. European Journal of Marketing, 55(4), 1227-1257. https://doi.org/10.1108/EJM-11-2018-0769

Khaw, K. W., Alnoor, A., Al-Abrrow, H., Tiberius, V., Ganesan, Y., & Atshan, N. A. (2023). Reactions towards organizational change: A systematic literature review. Current Psychology, 42(22), 19137-19160. https://doi.org/10.1007/s12144-022-03070-6

Kim, M., Choi, D., Guay, R. P., & Chen, A. (2023). How does fairness promote innovative behavior in organizational change?: The importance of social context. Applied Psychology, 73(3), 1233-1260. https://doi.org/10.1111/apps.12511

Li, M., Liu, W., Han, Y., & Zhang, P. (2016). Linking empowering leadership and change-oriented organizational citizenship behavior. Journal of Organizational Change Management, 29(5), 732-750. https://doi.org/10.1108/JOCM-02-2015-0032

Liao, P.-Y. (2023). Proactive personality, job crafting, and person-environment fit: Does job autonomy matter? Current Psychology, 42(22), 18959-18970. https:/

---

## [Editor Report · Decision Letter 1]

16 Oct 2024

Fairness Matters for Change: A Multilevel Study on Organizational Change Fairness, Proactive Motivation, and Change-Oriented OCB

PONE-D-24-21250R1

Dear Dr. liu,

We’re pleased to inform you that your manuscript has been judged scientifically suitable for publication and will be formally accepted for publication once it meets all outstanding technical requirements.

Kind regards,

Chunyu Zhang

Academic Editor

PLOS ONE
---

## [Editor Report · Acceptance letter]

20 Oct 2024

PONE-D-24-21250R1 

PLOS ONE

Dear Dr. Liu, 

I'm pleased to inform you that your manuscript has been deemed suitable for publication in PLOS ONE. Congratulations! Your manuscript is now being handed over to our production team.

Kind regards, 

on behalf of

Dr. Chunyu Zhang 

Academic Editor

PLOS ONE